# Advanced Glycations End Products in the Skin as Biomarkers of Cardiovascular Risk in Type 2 Diabetes

**DOI:** 10.3390/ijms23116234

**Published:** 2022-06-02

**Authors:** Alejandra Planas, Olga Simó-Servat, Cristina Hernández, Rafael Simó

**Affiliations:** 1Diabetes and Metabolism Research Unit, Vall d’Hebron Research Institute (VHIR), Vall d’Hebron University Hospital, Autonomous University of Barcelona, 08035 Barcelona, Spain; a.planas@vhebron.net (A.P.); olga.simo@vhir.org (O.S.-S.); cristina.hernandez@vhir.org (C.H.); 2CIBER de Diabetes y Enfermedades Metabólicas Asociadas (CIBERDEM), Spanish Institute of Health (ISCIII), 28029 Madrid, Spain

**Keywords:** cardiovascular disease, advanced glycation end products, diabetic retinopathy, cardiovascular disease biomarkers, type 2 diabetes, diabetic complications

## Abstract

The incidence and prevalence of diabetes are increasing worldwide, and cardiovascular disease (CVD) is the leading cause of death among subjects with type 2 diabetes (T2D). The assessment and stratification of cardiovascular risk in subjects with T2D is a challenge. Advanced glycation end products are heterogeneous molecules produced by non-enzymatic glycation of proteins, lipids, or nucleic acids. Accumulation of advanced glycation end products is increased in subjects with T2D and is considered to be one of the major pathogenic mechanism in developing complications in diabetes. Skin AGEs could be assessed by skin autofluorescence. This method has been validated and related to the presence of micro and macroangiopathy in individuals with type 2 diabetes. In this context, the aim of this review is to critically summarize current knowledge and scientific evidence on the relationship between skin AGEs and CVD in subjects with type 2 diabetes, with a brief reference to other diabetes-related complications.

## 1. Introduction

The incidence and prevalence of diabetes are increasing worldwide [1]. Diabetes is one of the leading causes of mortality and major morbidities, including cardiovascular disease (CVD), chronic kidney disease, and blindness [2]. Due to advances in healthcare and the widespread screening of serum glucose, the occurrence of complications has significantly decreased in recent years [3]. Even so, diabetes and its complications still rank as some of the most common causes of death and quality-of-life deterioration due to disease progression [4]. CVD is the leading cause of death among diabetic patients in whom adverse cardiovascular outcomes occur, which on average is 14.6 years earlier and with increased severity compared to individuals without diabetes [5]. People with type 2 diabetes (T2D) have a two-fold increased risk of developing CVD [5,6].

It is well known that chronic hyperglycemia is related with chronic complications of diabetes. However, two large studies revealed that tight glucose control slightly but not significantly reduced the risk of cardiovascular disease in either type 1 [7] or type 2 diabetes patients [8]. Furthermore, the exaggerated risk for CVD in this population is not fully explained by conventional risk factors such obesity, hyperglycemia, dyslipidemia, and hypertension, and in fact a substantial proportion of this risk remains unexplained [9,10]. Therefore, specific diabetes-related risk factors should be accounted for in assessments of excess risk for CVD, and the accumulation of advanced glycation end products (AGEs), heterogeneous compounds produced by the non-enzymatic reaction of glucose to proteins through the Maillard reaction, could be among them. In recent years, a simple and non-invasive method for AGE assessment through skin autofluorescence (SAF) has been developed. This method is based on specific fluorescence of certain skin AGEs, and validation studies have shown a strong correlation between SAF and the content of specific AGEs in skin biopsies [11,12].

In this context, the aim of this review is to critically summarize current knowledge and scientific evidence on the relationship between skin AGEs and CVD in subjects with type 2 diabetes, with a brief reference to other diabetes-related complications. 

## 2. Formation of AGEs and Physiopathology

AGEs are formed by the Maillard process, which is a non-enzymatic glycation of proteins, lipids, or nucleic acids. Protein glycation is mainly started when the carbonyl groups of reducing sugars, such as glucose, interact non-enzymatically with the reactive amino group of proteins, such as lysine or arginine residues. After that, this interaction forms an unstable aldimine compound, the Schiff base. The Schiff base can be rearranged to produce a stable Amadori product (for example HbA1c), which accumulates on proteins over a period of several weeks. The Amadori product undergoes oxidative degradation to generate highly reactive intermediate dicarbonyl compounds that interact again with free amino groups of proteins. Then, complex chemical reactions occur, and thus a highly heterogeneous, often fluorescent, insoluble, and irreversible group of AGEs is formed, which accumulates and damages long-lived proteins such as extracellular matrix collagen. In summary, in the Maillard process, there are early stage reactions that lead to the formation of early glycation adducts (such as HbA1c), and later-stage reactions subsequently form AGEs [13]. 

AGEs accumulate in the body during aging, but the degree of accumulation of AGEs is associated with increased production and decreased degradation and renal clearance. In patients with diabetes, chronic hyperglycemia accompanied by hyperlipidemia, oxidative/carbonyl stress, and, sometimes, decreased renal function leads to the accumulation of AGEs [14]. Accumulation of AGEs could be considered as one of the major pathogenic mechanisms resulting in end-organ damage in subjects with diabetes [15]. 

The formation and accumulation of AGEs can cause damage and may contribute to diabetic complications mainly by two pathways. First, cross-links can be formed with long-lived proteins in the body such as those constituting the extracellular matrix (ECM) and vascular basement membranes (BMs). These proteins are highly susceptible to AGE-modification. Functionally, AGE-mediated crosslinks in BM are known to cause reduced solubility and decreased enzymatic digestion [16]. Moreover, AGE formation has been shown to affect the three-dimensional nature of BM proteins, thereby causing structural and functional abnormalities. For example, AGE-modification of vitronectin, laminin, and collagen can seriously alter molecular charge characteristics, upset the ability to form precisely assembled matrix aggregates, and thus disrupt biological attachment sites that enable cells to adhere to their substrates [15]. Thus, the presence of AGE on vascular BM may have direct pathological consequences, particularly in diabetics, who have accelerated formation and accumulation of AGEs.

Second, AGEs can cause deleterious effects by the activation of receptors for AGEs (RAGEs). The most widely studied is RAGE, but other binding proteins include AGE receptors (Rs) 1, 2, and 3 (AGE-R1, AGE-R2, and AGE-R3/galactin-3, respectively), and the ezrin, radixin, and moesin (ERM) family [17]. 

RAGE is a member of the immunoglobulin superfamily of receptors. AGEs, by interacting with RAGE, trigger the activation of secondary messenger pathways such as protein kinase C. A crucial target of RAGE signaling is nuclear factor (NF)-KB, which is translocated to the nucleus where it increases transcription of a number of proteins, including endothelin-1, intercellular adhesion molecule-1, tissue factor, E-selectin, vascular endothelial growth factor (VEGF), and proinflammatory cytokines and mediators of oxidative stress [18,19]. All these molecular mediators are involved in the development of diabetic complications. The main mechanisms by which AGE accumulation participates in the development of complications in T2D are summarized in Figure 1.

Endothelial damage is a common feature in diabetic complications, and the increase of capillary permeability (or vascular leakage) is one of its hallmarks. In this regard, the activation of the ezrin, radixin, and moesin (ERM) complex deserves a brief comment. ERM includes membrane-associated proteins and acts as a cytoskeleton-membrane linker. ERM proteins present two conformations: an inactivated one, in which they are folded by an intramolecular interaction between the amino- and carboxyterminal domains; and an activated conformation, where the two domains separate, unmasking their binding sites. ERM protein activation in endothelial cells induces the cytoskeleton reorganization in stress fibers, leading to the disassembly of focal adhesions and the formation of paracellular gaps, which result in an increase of vascular permeability [20]. The activation of these proteins is induced by mediators involved in diabetic complications such as AGEs, oxidative stress, PKC activation, and TNF-α. It is known that the interaction between AGE and its receptor (RAGE) activates the MAPK and RhoA kinase signaling pathways, which are both able to induce moesin phosphorylation [21]. Furthermore, there is evidence that vascular leakage induced by AGEs and mediated by moesin phosphorylation also occurs in endothelial cells of brain and retina in murine models, and in human umbilical vein cell (HUVEC) cultures [21]. In short, AGE accumulation and the activation of RAGE cause moesin phosphorylation, which plays a key role in vascular leakage and endothelial dysfunction.

## 3. Assessment of AGEs

The plasmatic determination of AGEs, such us N-ε-carboxymethyl lysine (N-ε-CML) or pentosidine, have been proposed as biomarkers for diabetic complications. Several papers have shown that circulating levels of AGEs in patients with diabetes are associated with the progression of atherosclerosis [22], renal failure [23], or diabetic retinopathy (DR) [24]. However, there are also other studies that did not show the same association [25,26,27]. Circulating AGEs are rapidly broken down to AGE peptides or free AGEs, which are excreted by the kidney, thus having a fast turnover [28]. Moreover, biochemical and immunochemical assays for circulating AGE determinations are complex, time consuming, expensive, and of low reproducibility [29]. In addition, there is a significant variation with renal function. All these reasons limit their use in current clinical practice. 

Vlassara et al. [30] demonstrated that tobacco use and nutritional intake of AGE-rich meals (such as the modern western diet, where food is processed for safety, conservation, and the improvement of taste, flavor, and appearance) influences AGE accumulation. Moreover, cooking methods that utilize high temperature and low moisture increase the AGE content of food above the uncooked state [31]. Adherence to a Mediterranean diet (the pattern of which is based on foods with a low content of AGEs, such as vegetables, fruits, fish, whole grains, olive oil, and nuts) was inversely associated with SAF [32].

Serum AGEs do not necessary reflect tissue AGE levels. Since AGEs accumulate in long-lived proteins, it seems reasonable to assess AGEs in accessible tissues such as the skin, where long-lived proteins are present. Skin AGEs are mainly accumulated in collagen, which has a low turnover and represents the diabetic milieu influence over a longer time period than HbA1c; thus, skin AGEs may reflect the impact of both oxidative stress and a history of sustained hyperglycemic episodes [33]. The first evidence that accumulation of AGEs in skin tissue was related to the presence of micro and macrovascular complications in type 1 diabetes was in 1986 [33]. Some years later, the DCCT-EDIC sub study showed that skin AGEs levels measured in biopsy specimens were associated with the development and progression of diabetic complications in type 1 diabetes, even after adjustment for HbA1c [34]. Similar results were also reported in type 2 diabetes in the UKPDS [8]. 

Nevertheless, the assessment of AGEs in skin biopsy is not feasible in daily clinical practice. Based on specific fluorescence of some AGEs, a simple and non-invasive method for skin AGEs assessment has recently been developed through skin autofluorescence (SAF). Skin autofluorescence is measured using an autofluorescence reader (AGE Reader^TM^ device (DiagnOptics TechnologiesBV, Groningen, the Netherlands)), which illuminates 4 cm^2^ of the skin surface on the volar side of the forearm, guarded against surrounding light, and uses an excitation light source with a peak excitation of 370. Subsequently, the emitted fluorescence light (within the wavelength range of 420–600 nm) and the reflected excitation light (within the wavelength range of 300–420 nm) from the skin are measured with a spectrometer. SAF is calculated in arbitrary units (AUs) as the ratio between the emitted light and the reflected light, multiplied by 100. A series of three consecutive measurements are carried out, taking less than a minute [11]. Notably, it has been demonstrated that SAF has a strong correlation with the specific AGEs, such pentosidine, carboxymethyl-lysine, or carboxyethyl lysine content in skin biopsies [11,12]. 

## 4. SAF and Diabetic Microvascular Complications

It is well known that SAF values are related with the development of diabetic micro and macrovascular complications, and this is supported by multiple evidence, not only in cross-sectional studies [35,36,37,38,39,40,41] but also in prospective trials [42,43]. Wang et al. [44] recently published a large cross-sectional study comprising 825 subjects with type 2 diabetes showing that SAF is an independent predictor of T2D complications, including DR, diabetic kidney disease, diabetic cardiovascular disease, and diabetic peripheral neuropathy. Additionally, as the number of complications increases, the SAF value also increases. Hosseini et al. [45], in a systematic review and meta-analysis, suggested that SAF levels could be a predictor of chronic micro and macrovascular complications in DM.

In diabetic nephropathy, the majority of studies has reported a positive association between SAF and diabetic nephropathy [27,40], but some of them did not find this association [36,46]. It seems that in the kidney, activation of RAGE with AGEs may induce podocyte apoptosis and generation of monocyte chemoattractant peptide-1 and transforming growth factor-β, leading to albuminuria and glomerular sclerosis [47]. Moreover, in populations with end-stage renal disease, SAF is associated with cardiovascular events (CVE) and predicts mortality in subjects with and without diabetes [48,49,50]. Shardlow et al. [48] published a large study including 1707 subjects with chronic kidney disease (CKD) stage 3, with a follow up of 5 years in which fatal and non-fatal CVE were collected. The Kaplan–Meier analysis showed a progressive increase in CVE across tertiles of baseline SAF. Additionally, multivariable analysis identified SAF as an independent risk factor for time to first cardiovascular event in subjects with early stage 3 CKD. These findings have not only been seen in subjects with early stages of CKD, but also in patients with end-stage kidney disease. Furuya et al. [50] demonstrated that skin AGEs values were significantly higher in hemodialysis patients with de novo CVD in comparison with those patients without CVD. It is known that reduced nitric oxide production and/or its bioavailability is a common feature in high-risk patients such as diabetes, leading to endothelial dysfunction and CVD. AGEs can contribute to this alteration, in particular in the setting of CKD. In this regard, Ando et al. [51] found that (1) AGEs increase the level of an endogenous nitric oxide synthase inhibitor, asymmetric dimethylarginine, in endothelial cells; and (2) circulating levels of AGEs are correlated with serum asymmetric dimethylarginine and are inversely associated with endothelial function in diabetic patients with end-stage renal disease. These findings suggest that the link between AGE and asymmetric dimethylarginine could be a mediator involved in the high cardiovascular risk that present those patients with CKD. 

In the case of DR, evidence is controversial. Some studies reported a lack of association between DR and skin AGEs [36,52]. However, most recent studies have found a clear independent correlation with development of retinopathy and its severity [38,46,53,54,55]. Interestingly, Takayanagi et al. [55] demonstrated that skin AGEs are not only related with the presence and severity of DR but also with the progression of DR. It is believed that the association between skin AGEs and DR is due to the important role of AGEs in the oxidative stress-induced apoptosis of the retinal pericytes [56]. It is known that AGEs can induce intrinsic signaling pathways mediated mainly through RAGEs expressed on the membrane of pericytes, leading to apoptosis [57]. Since pericyte function is the main regulator of the basement membrane at the blood retinal barrier [58], selective pericyte loss leads to disruption of the blood retinal barrier and the development of DR [59]. In addition, AGE accumulation upregulates VEGF, a major mediator of diabetic macular edema and proliferative DR [60,61]. Lu et al. [61] demonstrated that AGEs can stimulate the expression of VEGF in rat and rabbit retina; to examine whether AGEs increase retinal VEGF mRNA levels in vivo, AGEs were injected into the vitreous of rat and rabbit eyes, and in situ hybridization studies and Northern blot analyses were completed. Rat retinal VEGF mRNA levels were increased in the ganglion, inner nuclear, proximal photoreceptor, and retinal pigment epithelial and choroidal layers of the AGE-injected rat and rabbit eyes. Moreover, Northern blot analyses of rabbit neurosensory retina identified a 4.8-fold increase in VEGF mRNA levels in the AGE-injected eyes. These data provide a potential mechanistic link between hyperglycemia, VEGF, and DR.

The association between diabetic neuropathy (DN) and SAF has been reviewed recently by Papachristou et al. [62], and the association is quite unanimously agreed upon [63,64]. Most evidence shows that increasing SAF levels predicts the development of DN [43,64,65]. In addition, increases in skin AGEs may precede small sudomotor dysfunction and altered vibration perception threshold [64,66]. It is believed that the accumulation of AGEs in the peripheral nerves leads to the enhancement of reactive oxygen species, which promotes neural inflammation and impairs axonal transport. These perturbations, along with direct neuronal toxicity from intracellular sorbitol accumulation (due to hyperglycemia), culminate in DN [62]. Nevertheless, it should be noted that published studies are heterogeneous, including populations with different diabetes type, different SAF cut-off values, and different methods of DN assessment, so this evidence must be taken with caution. 

## 5. SAF and Diabetic Macrovascular Complications

Subjects with diabetes presented an increased risk for myocardial infarction and stroke caused by vascular occlusion and are more likely to develop serious cardiovascular and cerebrovascular disease than non-diabetic subjects [67,68]. The vascular occlusion process is pathophysiological and characterized by plaque formation. The interactions between cytokines, growth factors, and the different vessel wall cell types that contribute to atherogenesis are extremely complex and multifactorial. Atheromatous plaque formation in subjects with diabetes is practically the same from that occurring in non-diabetic subjects, although the distribution of plaques may be different, and diabetic lesions characteristically show a higher tendency for focal medial calcification [69]. AGEs have been accepted as having a key role in the formation and acceleration of atherosclerotic lesions, even in normoglycemic patients, but especially in diabetics [15].

The assessment and stratification of cardiovascular risk in subjects with T2D is a challenge. The UKPDS risk score is still one of the most used tools to give cardiovascular risk estimates in people type 2 diabetes [70]. Lutgers et al. demonstrated that SAF provides additional information to the UKPDS risk score for the estimation of cardiovascular prognosis in T2D [71]. In addition, there is emerging evidence indicating that SAF is an important biomarker not only of the presence of cardiovascular disease but also of their outcomes [72,73]. 

AGEs may contribute to cardiovascular events and cardiovascular mortality by three well-established pathophysiological mechanisms: (1) AGEs can affect the physiological properties of cardiac proteins in the extracellular matrix by creating cross-links, which provoke decreased flexibility of the matrix proteins and produce stiffness in vascular walls [74]; (2) AGEs induce endothelin-1 production [75] and reduce nitric oxide [76] at the vascular level, thus resulting in vasoconstriction and the loss of vascular compliance; and (3) AGEs can cause multiple vascular and myocardial changes through the interaction with RAGEs, leading to atherosclerosis, thrombosis, and vasoconstriction [77]. It should be noted that RAGEs mediate the induction of fibrosis through the increase of TGF-β [78] and influence calcium metabolism in cardiac myocytes [79]. 

### 5.1. SAF and Subclinical Cardiovascular Disease

It is well established that SAF is a good predictor of subclinical cardiovascular disease in patients with and without diabetes (Table 1). 

Arterial stiffness is associated with the prevalence of CVD and predicts future cardiovascular events in healthy and high-risk patients [6]. The main components of the extracellular matrix within the arterial wall are type I collagen, type III collagen, and elastin. AGE accumulation leads to quantitative and qualitative alterations of collagens and elastin, which could contribute to the decreased elastic properties of the vessels, thereby playing a role in arterial stiffness [87]. SAF is strongly correlated with pulse wave velocity, brachial and aortic augmentation indices, and ankle-brachial index, all of them markers of arterial stiffness [72]. Birukov et al. [84] recently investigated the relationships between SAF and vascular stiffness in a large study performed in diabetic and non-diabetic populations. These authors concluded that SAF might be involved in vascular stiffening independently of cardiometabolic risk factors, and it could be a rapid and non-invasive method for the assessment of macrovascular disease progression across all glycemic strata [84]. However, Osawa et al. [82], in a smaller study including only subjects with type 2 diabetes, showed that SAF was significantly associated with C-IMT and pulse wave velocity (PWv), but it was not an independent determinant of C-IMT and PWv after adjustment for confounders [82].

Carotid intima–media thickness (IMT) is a useful marker of the progression of atherosclerosis and is an excellent predictor of cardiovascular events. SAF was an independent determinant of max-IMT (R = 0.45, β = 0.425, *p* < 0.01) in a small study with T2D subjects [37]. Regarding SAF and atherosclerosis, a large study comprising 1013 subjects with T2D showed a clear association between SAF and lower-extremity atherosclerotic disease (LEAD) assessed by ultrasound [86].

Basic research has shown that the interaction of AGEs with RAGE in atherosclerotic plaques trigger the production of inflammatory mediators, which lead to plaques more vulnerable to rupture [88]. In addition, data regarding the important role of oxidative stress on endothelial dysfunction and coronary artery disease are extensive [89]. However, most markers for oxidative stress are not readily available for clinical practice. It is well known that AGEs, by interacting with their own receptor RAGE, can induce intracellular signaling that leads to enhanced oxidative stress [14]. Moreover, skin AGEs are stable and could be non-invasively assessed, thus serving as a reliable biomarker of cardiovascular disease.

Coronary artery calcification score (CACs) is a common feature in advanced atherosclerosis and a powerful predictor of future cardiovascular events such as myocardial infarction [6]. Our group has recently published a study comprising 156 subjects with T2D and 52 controls, and we have demonstrated that SAF is a good and independent predictor of CACs ≥ 400 with OR 2.04 (CI 95% 1.07–3.88), *p* = 0.033, with area under the ROC curve of 0.77 (CI 95% 0.70–0.84) [85].

A recent meta-analysis and systematic review on the association of arterial stiffness measured by PWv and atherosclerosis measured by carotid IMC with SAF has been published [72]. The authors concluded that a positive weak association of PWv and carotid IMC with SAF does exist. 

These findings support the concept that AGEs and their receptor system (RAGE) play an important role in the impairment of vascular function. Thus, AGEs are not only markers of “metabolic memory” in diabetic subjects but also have an important pathogenic role both in endothelial dysfunction and in the atherosclerotic process.

### 5.2. SAF and Cardiovascular Disease and Mortality

There is increasing evidence that SAF is a robust predictor of cardiovascular events and cardiovascular death in subjects with T2D. In Table 2 we summarize this best evidence. 

In a multicenter cross-sectional study comprising more than 500 T2D subjects, Noordzij et al. [36] showed that SAF values were higher when a greater number of diabetic complications was present. In addition, these authors observed that SAF was associated with the presence of macrovascular complications in patients with diabetes, independently of classical risk factors. 

Mulder et al. [95] showed that SAF is elevated in acute ST-elevation myocardial infarction compared with healthy controls. In addition, higher values of SAF were related with more risk to die or to present a new myocardial infarction or heart failure in the following year. 

Skin AGEs are not only associated with CVD and are useful as predictors of cardiac events but are also associated with peripheral artery disease and can be considered as a useful biomarker to predict amputations in these patients. In this regard, de Vos et al. [90] demonstrated in a prospective study (5-year follow-up) comprising 252 subjects with peripheral artery disease that SAF values were a strong predictor of amputation, with a hazard ratio of 3.05 (CI 95% confidence interval [CI], 1.87–4.96); *p* < 0.0001).

Meerwaldt et al. [42], using a cohort of 69 T2D subjects with a follow-up of 5 years, were the first to show that SAF was strongly associated with cardiac mortality (OR: 2.9; CI 95% 1.3–4.4). Yozgatli et al. [91], in a large and multicentric study comprising 563 subjects with T2D with a follow up of 5 years, showed that SAF was a significant predictor of fatal and non-fatal macrovascular events (HR 1.28 CI 95% 1.03–1.6, *p* < 0.001). In addition, participants in the highest SAF tertile developed almost twice as many macrovascular events compared with those in the lowest tertile. Interestingly, these authors found that whereas SAF was associated with development of macrovascular events in people with type 2 diabetes, HbA_1c_ was associated with the development of microvascular complications.

Cavero-Redondo et al. [73] published some years ago a systematic review and meta-analysis about SAF as a predictor of cardiovascular and all-cause of mortality in high risk subjects with renal or cardiovascular disease. Ten studies were included, but only two with diabetic populations. They concluded that higher SAF levels were significantly associated with higher pooled risk estimates for cardiovascular mortality (HR: 2.06; 95% CI, 1.58–2.67) and all cause of mortality (HR: 1.91; 95% CI, 1.42–2.56). Therefore, SAF level could be considered a predictor of all-cause mortality and cardiovascular mortality in subjects with high risk with previous cardiovascular and kidney disease. 

A recent article by Boersma et al. [93] explored the relation between SAF levels and the development of type 2 diabetes, cardiovascular disease, and mortality, and it evaluated if elevated SAF values may predict the development of CVD and mortality in individuals with T2D. A total of 2349 subjects with T2D was included; 1318 reported a previous diagnosis of T2D (median duration of the disease of 5 years), while the rest of the included subjects were “new” cases of diabetics since the diagnosis was performed at baseline due to altered fast glycaemia or an HbA1c out of range. They followed up these patients a mean of 3.7 years and collected new CV events. They observed that individuals with “new” T2D had lower SAF values than those with known type 2 diabetes, reflecting the longer period of exposure to elevated glucose levels. In addition, SAF was significantly and independently associated with the combined outcome of new CV events and mortality in T2D subjects (OR 2.59, 95% CI 2.10–3.20, *p* < 0.001). 

Recently, Chen et al. [96] published a meta-analysis evaluating the prospective association between skin AGEs and major adverse cardiovascular events (MACEs). They concluded that the higher levels of SAF are significantly correlated with a higher pooled risk of MACE.

We have recently published a prospective case-control study with 4.35 year of follow-up in which 187 subjects with T2D without any apparent cardiovascular disease and 57 healthy age-matched controls were included. We found that SAF together with DR were powerful predictors of CV events, and the higher values of SAF were independently associated with the presence of CV events (HR 4.68 CI 95% 1.83–11.96, *p* = 0.001) [94]. 

These findings support the clinical utility of SAF to support risk assessment for CVD and mortality, both in the general population and in people with type 2 diabetes

## 6. Conclusions

AGE accumulation has been demonstrated to play a pathophysiological role in the development of chronic complications in diabetes. Moreover, SAF assessment has been revealed to be an important biomarker of AGE burden and represents a more long-term memory of cumulative metabolic stress than does HBA1c and other conventional risk factors. As mentioned above, there is accumulating evidence to show the clinical utility of the measurement of SAF for evaluating vascular risk in diabetes patients. We believe that SAF could be a useful and simple tool, and clinicians should consider the level of SAF for assessment of cardiovascular risk in subjects with type 2 diabetes. However, more research is needed to establish an optimal and reliable cut-off of SAF for different populations to help the clinicians to make clinical decisions. 

## Figures and Tables

**Figure 1 ijms-23-06234-f001:**
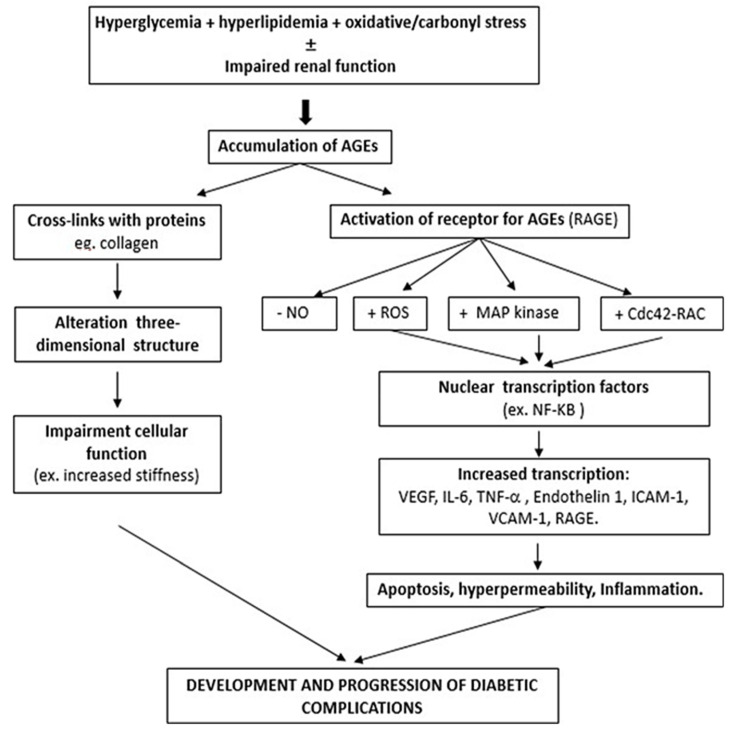
Multi-pathway contribution of AGEs to diabetic complications. Accumulation of advanced glycation end product (AGE) may result from hyperglycemia, hyperlipidemia, and oxidative stress, with or without impaired renal function. AGEs can form cross-links with proteins that affect the three-dimensional structure and thereby the functions of these proteins, and they can also cause deleterious effects by the activation of receptors for AGEs (RAGEs), which in turn can lead to activation of second messengers and transcription factors that up-regulate pro-inflammatory cytokines and mediators of oxidative stress. These effects modify pathways which contribute to the development and progression of diabetic complications. NO, nitric oxide; ROS, reactive oxygen species; MAP, mitogen-activated protein; Cdc42, cell division cycle 42 protein; NF-KB, nuclear factor kappa-light-chain-enhancer of activated B cells; VEGF, vascular endothelial growth factor; TNF-α, tumor necrosis factor α; ICAM-1, intercellular adhesion molecule-1, VCAM-1 Vascular cell adhesion protein 1.

**Table 1 ijms-23-06234-t001:** SAF as a biomarker of the presence of subclinical cardiovascular disease.

First Author (Year)	Participants and Diabetes Type	Measurement	Main Findings
Temma (2015) [37]	61 T2D	C-IMT	SAF well correlated with the degree of max-IMT of the carotid artery.
Hangai (2016) [27]	122 T2D	baPWV; C-IMT; CACs	SAF positively correlated with CACs. Stronger with CACs than either PWV or IMT.
Fujino (2018) [80]	108 (50% T2D)	Coronary plaques assessed by OCT.	SAF positively associated with more vulnerable and calcified plaques.
Ninomiya (2018) [81]	140 (T1D and T2D)	Subclinical atherosclerosis: FMV, IMT, baPWV	SAF is an independent determinant of brachial FMD (indicator of endothelial dysfunction), and SAF is associated with IMT and baPWV (markers of early-stage atherosclerosis).
Yoshioka (2018) [46]	162 T2D and 42 controls	C-IMT	SAF was an independent determinant of max-IMT (early-stage atherosclerosis).
Osawa (2018) [82]	193 T2D and 24 controls	C-IMT, ankle-brachial index, baPWV	SAF was significantly associated with C-IMT and baPWV but was not an independent determinant of C-IMT and baPWV after adjustment for confounders.
Jujić (2019) [83]	496 (10% T2D)	Carotid ultrasound. (TPA)	SAF is associated with the degree of atherosclerosis. A 1 SD increment in SAF is associated with increased atherosclerotic burden (TPA).
Sánchez (2019) [25]	2568 (non-diabetic subjects)	TPA (vascular carotid and femoral ultrasound)	SAF is associated with increased atherosclerotic burden (the presence of plaque, number of affected territories, and TPA).
Birukov (2021) [84]	1348 (T2D and non-diabetic subjects)	Vascular stiffness: carotid-femoral and aortic PWV and brachial and aortic augmentation indices.	SAF is positively associated with measures of arterial stiffness, independent of potential cardiometabolic confounders and glycemic status.
Planas (2021) [85]	156 T2D and 52 non-diabetic subjects.	Coronary atherosclerosis assessed by CACs.	SAF is a good and independent predictor of CACs ≥ 400.
Ying (2021) [86]	1013 T2D	LEAD (color doppler ultrasonography).	SAF is associated with the presence of lower extremity atherosclerosis.

T2D: Subjects with type 2 diabetes; TD1: subjects with type 1 diabetes; C-IMT: carotid intima–media thickness; baPWV: brachial-ankle pulse wave velocity; PWV: pulse wave velocity; CACs: coronary artery calcium score; FMV: flow-mediated vasodilation; SD: standard deviation; TPA: total plaque area; LEAD: lower-extremity atherosclerotic disease.

**Table 2 ijms-23-06234-t002:** SAF as a biomarker of cardiovascular outcomes.

First Author (Year)	Participants and Diabetes Type	Outcome	Follow Up	Main Findings
Meerwaldt (2007) [42]	69 T2D, 48 T1D, and 43 controls	CV mortality	5 years	SAF strongly associated with CV mortality. OR 2.9 CI 95% 1.3–4.4 for T2D, and OR 2.0 CI 95% 1.3–2.7 for T1D.
Tanaka (2011) [40]	130 T2D	Ancient macrovascular complications	Cross sectional	SAF associated with macrovascular complications (OR 7.25 CI 95% 2.22–23.7).
Noordzij (2012) [36]	563 T2D	Ancient macrovascular complications	Cross sectional	SAF was associated with macrovascular complications.
De Vos (2015) [90]	267 (10% T2D)	New amputations in patients with PAD	5.3 years	SAF predicts amputations in patients with PAD independent of diabetes. HR 2.72 (CI 95% 1.38–1.539) per unit of SAF for amputation.
Furuya (2015) [50]	64 subjects with CKD in hemodialysis (56.3% subjects with diabetes)	New CV events	3 years	SAF is significantly associated with incidence of new CV event OR 2.96 CI 95% 1.26–8.16
Siriopol (2015) [49]	304 dialysis subjects (18.4% diabetic subjects)	CV mortality, sepsis-related mortality, other causes of mortality	2.5 years	SAF is associated in all-cause (HR 2.09 CI 95% 1.24–3.59) and sepsis-related mortality (HR 3.44 CI 95% 1.59–7.42).
Yozgatli (2018) [91]	563 T2D	New CV events and microvascular complications	5 years	SAF is a significant predictor of fatal and non-fatal CV events (HR 1.53 CI 95% 1.24–1.48 per unit of SAF in the development of CV events.
Kunimoto (2021) [92]	204 subjects with heart failure and CVD (30% T2D)	Major CV event (all cause of mortality + unplanned hospitalization for heart failure)	1.6 years	Higher SAF levels are significantly and independently associated with major CV events. SAF was associated with major CV adverse event (OR 2 CI 95% 1.41–2.78, *p* < 0.01).
Boersma (2021) [93]	1318 T2D 1031 new T2D	New CV events	3.7 years	SAF is significantly and independently associated with the new CV event and mortality in people with T2D (OR 2.59 CI 95% 2.1–3.2).
Planas (2021) [94]	187 T2D and 57 controls	First CV event	4.35 years	Higher values of SAF are predictors of new CV events (HR 4.68 CI 95% 1.83–11.96).

T2D: subjects with type 2 diabetes; TD1: subjects with type 1 diabetes; CV: cardiovascular; PAD: peripheral artery disease; OR: odds ratio; CI: confidence interval; HR: hazard ratio.

## Data Availability

The data presented in this study are available upon request from the corresponding author.

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
