# Peer review of "Advanced Glycations End Products in the Skin as Biomarkers of Cardiovascular Risk in Type 2 Diabetes"

_ijms, 2022, doi:10.3390/ijms23116234_

Round 1
Reviewer 1 Report
Here, Planas and colleagues review the role of Advanced Glycations End Products (AGEs) in type 2 diabetes (T2D). In particular, the authors focus on AGEs measured as skin autofluorescence (SAF) and on their role as biomarkers of cardiovascular risk in T2D. The topic is worth of interest and the manuscript is clear and well written. I have only some minor concerns:
-The authors could briefly discuss this study in their manuscript (PMID: 35260315)
-Lines 70-71: in my opinion, the sentence is too “strong”; the authors could rephrase or expand it by adding more supporting references.
-Line 89: spell out “Ig”
-In figure 1: I don’t understand what “±” impaired renal function refers to. Moreover, the authors should expand the figure 1 legend, which only consists of the title and abbreviations.
-Line 111: who is activated? Endothelial cells or ERM proteins? Please rephrase
-Line 178: spell out “CVE”
-Spell out in the text “PWv”
Author Response
Here, Planas and colleagues review the role of Advanced Glycations End Products (AGEs) in type 2 diabetes (T2D). In particular, the authors focus on AGEs measured as skin autofluorescence (SAF) and on their role as biomarkers of cardiovascular risk in T2D. The topic is worth of interest and the manuscript is clear and well written. I have only some minor concerns:
ANSWER: Many thanks for the revision process and the kind comments on our paper.
- The authors could briefly discuss this study in their manuscript (PMID: 35260315)
This paper has been briefly commented and included in the reference list.
- Lines 70-71: in my opinion, the sentence is too “strong”; the authors could rephrase or expand it by adding more supporting references.
ANSWER: This sentence has been toned down as suggested.
- Line 89: spell out “Ig”
ANSWER: Done!
- In figure 1: I don’t understand what “±” impaired renal function refers to. Moreover, the authors should expand the figure 1 legend, which only consists of the title and abbreviations.
ANSWER: The symbol refers to impaired kidney function, that may or may not be present. So, the accumulation of AGEs could happen with or without the impairment of kidney function. This has been further clarified in the expanded figure legend.
- Line 111: who is activated? Endothelial cells or ERM proteins? Please rephrase
ANSWER: This has been clarified in the revised manuscript. Thank you!
- Line 178: spell out “CVE”
ANSWER: Done!
- Spell out in the text “PWv”
ANSWER: Done!
Reviewer 2 Report
In this review the authors critically summarize the current scientific knowledge and evidence on the relationship between cutaneous AGEs and CVD in subjects with type 2 diabetes.
The manuscript is interesting and well organized.
Perhaps it would be helpful if the authors add a grafical abstract so the results are immediate.
Are there studies in the literature that highlight the role of the microbiota in type 2 diabetes?
Food can influence? do the authors have any indications?
Author Response
In this review the authors critically summarize the current scientific knowledge and evidence on the relationship between cutaneous AGEs and CVD in subjects with type 2 diabetes.
The manuscript is interesting and well organized.
Perhaps it would be helpful if the authors add a grafical abstract so the results are immediate.
ANSWER: Thank you for the revision process and the kind comments on our paper. Since this is a review a graphical abstract is not recommended. However, we can prepare it if you find it necessary.
Are there studies in the literature that highlight the role of the microbiota in type 2 diabetes?
ANSWER: Although microbiota play an important role in pathophysiology of type 2 diabetes, we are unaware of relevant publications on its participation in AGEs accumulation.
Food can influence? do the authors have any indications?
ANSWER: The referee is right indicating that type of diet could influence AGEs generation and accumulation. A brief paragraph with appropriate references on this issue has been added to the revised manuscript (2nd paragraph: “Assessment of AGEs”)
This manuscript is a resubmission of an earlier submission. The following is a list of the peer review reports and author responses from that submission.